# Multiple antiferromagnetic phases and magnetic anisotropy in exfoliated CrBr$_3$ multilayers

Fengrui Yao [1,2] ✉, Volodymyr Multian[1,2,3], Zhe Wang [4] ✉, Nicolas Ubrig [1,2], Jérémie Teyssier [1,2], Fan Wu [1,2], Enrico Giannini [1], Marco Gibertini [5,6], Ignacio Gutiérrez-Lezama [1,2] & Alberto F. Morpurgo [1,2] ✉

In twisted two-dimensional (2D) magnets, the stacking dependence of the magnetic exchange interaction can lead to regions of ferromagnetic and antiferromagnetic interlayer order, separated by non-collinear, skyrmion-like spin textures. Recent experimental searches for these textures have focused on CrI$_3$, known to exhibit either ferromagnetic or antiferromagnetic interlayer order, depending on layer stacking. However, the very strong uniaxial anisotropy of CrI$_3$ disfavors smooth non-collinear phases in twisted bilayers. Here, we report the experimental observation of three distinct magnetic phases—one ferromagnetic and two antiferromagnetic—in exfoliated CrBr$_3$ multilayers, and reveal that the uniaxial anisotropy is significantly smaller than in CrI$_3$. These results are obtained by magnetoconductance measurements on CrBr$_3$ tunnel barriers and Raman spectroscopy, in conjunction with density functional theory calculations, which enable us to identify the stackings responsible for the different interlayer magnetic couplings. The detection of all locally stable magnetic states predicted to exist in CrBr$_3$ and the excellent agreement found between theory and experiments, provide complete information on the stacking-dependent interlayer exchange energy and establish twisted bilayer CrBr$_3$ as an ideal system to deterministically create non-collinear magnetic phases.

The emergence of smooth non-collinear magnetic phases in twisted bilayers of two-dimensional (2D) magnetic semiconductors relies on the different roles of intra and interlayer exchange interaction, and depends crucially on the strength of uniaxial magnetic anisotropy[1–6]. Since 2D magnetic semiconductors are formed by covalently bonded layers held together by weak van der Waals forces[7–12], the intralayer exchange is relatively strong and drives long-range magnetic ordering if magnetic anisotropy is also strong enough. Interlayer exchange is much weaker but has a key role, especially in 2D magnets whose spins point in the same direction within each layer, because it determines whether the system is a ferromagnet or a layered antiferromagnet[13–19]. As the strength and sign of interlayer exchange vary rapidly with atomic distances, whether interlayer coupling is ferromagnetic (FM) or antiferromagnetic (AFM) depends critically on

[1]Department of Quantum Matter Physics, University of Geneva, 24 Quai Ernest Ansermet, CH-1211 Geneva, Switzerland. [2]Department of Applied Physics, University of Geneva, 24 Quai Ernest Ansermet, CH-1211 Geneva, Switzerland. [3]Advanced Materials Nonlinear Optical Diagnostics lab, Institute of Physics, NAS of Ukraine, 46 Nauky pr., 03028 Kyiv, Ukraine. [4]MOE Key Laboratory for Nonequilibrium Synthesis and Modulation of Condensed Matter, Shaanxi Province Key Laboratory of Advanced Materials and Mesoscopic Physics, School of Physics, Xi'an Jiaotong University, Xi'an 710049, China. [5]Dipartimento di Scienze Fisiche, Informatiche e Matematiche, University of Modena and Reggio Emilia, IT-41125 Modena, Italy. [6]Centro S3, CNR-Istituto Nanoscienze, IT-41125 Modena, Italy. ✉e-mail: fengrui.yao@unige.ch; zhe.wang@xjtu.edu.cn; alberto.morpurgo@unige.ch

layer stacking[20–26]. It follows that in layers twisted with a small relative angle, the resulting moiré pattern causes the interlayer coupling to vary periodically in space and creates a lattice of relatively large islands, whose magnetic order is determined by the corresponding local atomic stacking. Non-collinear magnetic phases emerge when the moiré periodicity induces alternating FM and AFM islands, and the uniaxial anisotropy−while being sufficiently strong to stabilize long-range order within one layer−is not so strong to prevent smooth canting of the spins in the regions between the islands[1–6].

The Chromium trihalides (CrX$_3$; X = I, Br, Cl)[27–40], with ferromagnetically aligned spin within individual layers and stacking-dependent interlayer exchange interaction, offer ideal conditions to search for non-collinear moiré magnetic phases. That is why recent pioneering experiments have focused on twisted bilayer CrI$_3$[41,42]. However, since Iodine has the largest atomic number and therefore the strongest spin-orbit interaction[43], the very large uniaxial anisotropy of CrI$_3$ makes twisted bilayer of this compound not optimal for the stabilization of non-collinear phases. In the opposite limit, in CrCl$_3$, spin-orbit interaction is weak and experiments have established that the magnetic anisotropy is correspondingly weak, causing the ferromagnetic transition in monolayers to be of the Kosterlitz-Thouless type, without truly long-range order[44]. It should therefore be hoped that CrBr$_3$ may be suitable to engineer non-collinear magnetic phases in twisted bilayers, because its uniaxial anisotropy−while being sufficiently large to ensure ferromagnetic long-range order in monolayers[38]−is expected to be much weaker than in CrI$_3$. So far, however, this remains unexplored experimentally and, moreover, further progress is hampered because only FM coupling has been reported in exfoliated CrBr$_3$ multilayers[32,38–40]. To exploit the potential of CrBr$_3$ for the search of non-collinear magnetic phases in twisted bilayers it is therefore essential to demonstrate structures with both FM and AFM interlayer coupling in exfoliated layers, to fully characterize their interlayer magnetic exchange interaction, and to establish that the strength of magnetic anisotropy is indeed sizably smaller than in CrI$_3$.

Here, we report the observation of exfoliated CrBr$_3$ multilayers with three distinct magnetic interlayer couplings−and correspondingly distinct magnetic orders−which we associate to different stackings of the constituent CrBr$_3$ monolayers. We detect these different magnetic states by performing magnetoconductance measurements on tunnel junctions realized with exfoliated CrBr$_3$ multilayers that are found to include parts with different layer stackings, resulting in different interlayer exchange couplings. In particular, we find that−besides the FM interlayer coupling responsible for bulk ferromagnetism−two distinct AFM states are present. One of these AFM states appears to be the same observed in films grown by molecular beam epitaxy[45] and the other had not been observed earlier. Multilayers exhibiting different magnetic states are fingerprinted by the splitting of specific Raman modes, which enables us to establish the symmetry of the layer stackings corresponding to the different magnetic phases. Magnetoconductance measurements performed on the AFM multilayers also enable us to determine the magnetic anisotropy of CrBr$_3$ –approximately four times weaker than in CrI$_3$– and the full phase diagram. We find that the critical temperatures of all stacking-dependent magnetic phases are very close, as expected for 2D magnets in which the interlayer coupling is much weaker than the intralayer one. Our experimental results are in full agreement with the density functional theory calculations of ref. 24 (whose basic aspects are summarized in Fig. 1), and represent the first observation of all predicted magnetic states of CrX$_3$ multilayers (in all CrX$_3$ only two of the three predicted locally stable configurations had been reported experimentally). These results provide all the needed information to engineer and analyze the magnetic phases of twisted bilayer CrBr$_3$.

## Results

To illustrate how tunneling magnetoconductance measurements are employed to determine the interlayer magnetic coupling (or the magnetic state) of thin 2D magnetic semiconductors, we start by discussing the known magnetoconductance of devices realized on multilayers of either FM CrBr$_3$[32,39,40] or layered AFM CrI$_3$[28–31] (see Fig. 2a for the device schematics). Electron transport in these devices is phenomenologically understood in terms of Fowler-Northeim (FN) tunneling[28–31,40], with the electric field generated by the applied bias that tilts the conduction band in the barrier, causing an exponential increase in tunneling probability (Fig. 2b). The process results in strongly non-linear I-V curves (Fig. 2c), such that $\ln(I/V^2)$ is proportional to $1/V$ (Fig. 2d). A finite magnetoconductance ($\delta G(H, T) = (G(H,T) - G_0(T))/G_0(T)$, where $G(H,T)$ is the conductance measured at magnetic field $H$ and temperature $T$ and $G_0(T) = G(H = 0,T)$ occurs because the magnetic state of the material determines the height of the tunnel barrier[28–31,40]. The magnetoconductance, therefore, exhibits a characteristic evolution with $H$ and $T$ that is different for FM and layered AFM barriers.

For a FM barrier, the magnetoconductance is small at low $T$ (Fig. 2e), because the spins are already nearly fully polarized for $H = 0$, and the magnetic state remains virtually unchanged when a finite $H$ is applied[40]. Characteristic "lobes" in the magnetoconductance appear in the critical region for $T \sim T_C$, due to the divergence of the magnetic susceptibility near the Curie temperature (Fig. 2f), such that the application of an even small magnetic field causes large changes in magnetization[40]. In a strongly anisotropic layered antiferromagnet (Fig. 2g, h), instead, the magnetoconductance is large at low temperatures and exhibits two characteristic sharp jumps at a material-dependent field and twice that value (0.9 T and 1.8 T in CrI$_3$[28–31]). The jumps originate from flipping the magnetization of the outer layers in the barrier (at 0.9 T) and of the inner ones (at 1.8 T), with the value of 0.9 T providing a direct measure of the strength of interlayer exchange. Importantly, the sequence of jumps differs for bi-, tri-, and thicker layers: bi- and tri-layer show only one jump at 0.9 T or 1.8 T, respectively; four layers (4 L) or thicker layers show two jumps at 0.9 T and 1.8 T[28–31]. Therefore, magnetoconductance measurements on magnetic tunnel barriers indicate unambiguously whether the interlayer coupling is FM or AFM, and for antiferromagnets, provide information about the number of layers.

### Antiferromagnetic phases in CrBr$_3$

One of our key experimental observations comes from magnetoconductance measurements on CrBr$_3$ tunnel barriers realized with multilayers exfoliated from crystals in which Raman spectra show an additional peak of sizable magnitude (at -161 cm$^{-1}$; see Supplementary Fig. 1 for detail), which we attribute to the presence of an allotrope of CrBr$_3$ different from the known thermodynamically stable structure (due to a different stacking of the CrBr$_3$ layers[46,47]). Specifically, the data shown in Fig. 2e, f−with small and featureless magnetoconductance at low $T$−are characteristic of CrBr$_3$ tunnel barriers realized with thin multilayers with layers fully stacked as in the FM state of the material (as discussed in our earlier work[40]). In several other tunnel barriers, however, the magnetoconductance is different, as illustrated by five representative devices in Fig. 3a: it is much larger and exhibits sharp jumps. The jumps occur at a few different specific values of magnetic field (as indicated by the vertical gray dashed lines) for all the measured samples. The analysis of these jumps provides clear information about the different types of naturally occurring interlayer couplings between adjacent CrBr$_3$ layers, depending on their stacking.

We observed jumps in the magnetoconductance of 12 (out of 20) different CrBr$_3$ samples, with thicknesses ranging from 2.8 to 20 nm (Supplementary Fig. 2). Figure 3b summarizes the magnetic field values at which the jumps occur: 0.55 T and twice this value (1.1 T), and 0.2 T and twice this value (0.4 T). Finding jumps reproducibly

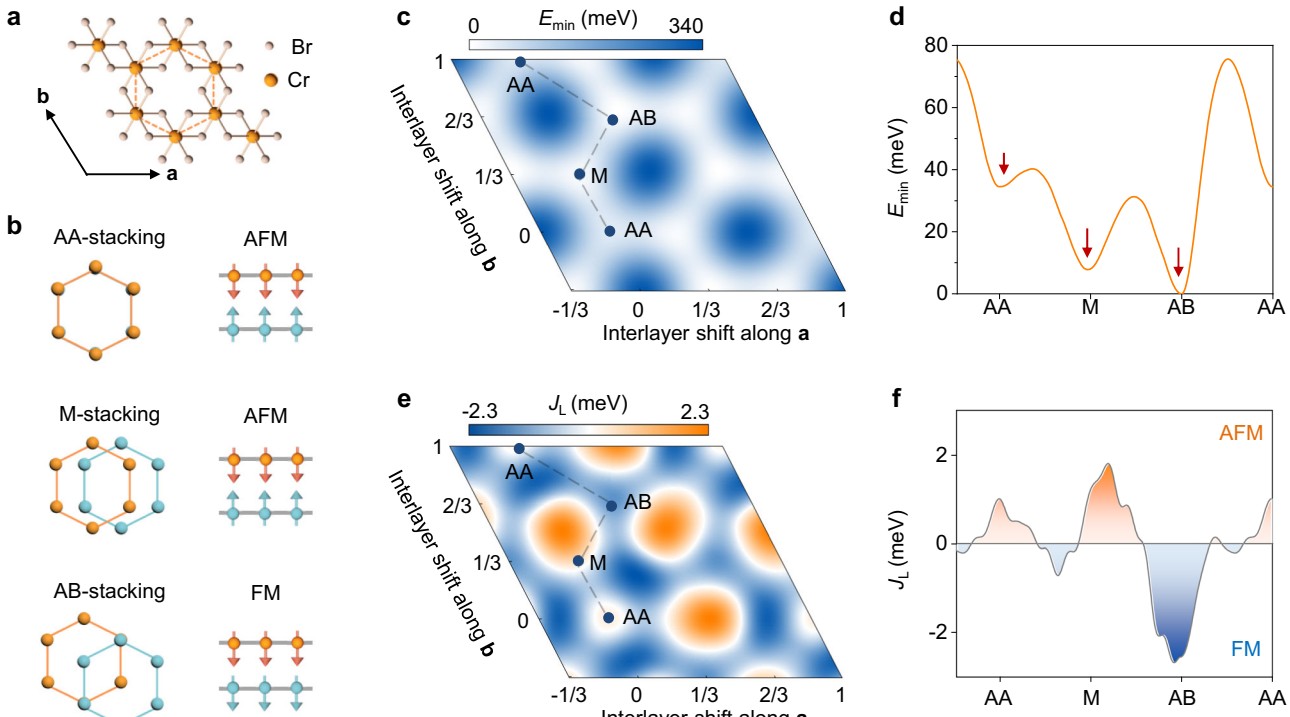

**Fig. 1 | Total and interlayer exchange energy of bilayer CrBr₃ as predicted by first-principles calculations.** **a** Top view of monolayer $CrBr_3$, where the Cr atoms (orange balls) form a honeycomb lattice and lie inside edge-sharing octahedra formed by the Br atoms (pink balls; **a** and **b** are the two primitive lattice vectors of the unit cell). **b** Atomic arrangement and interlayer magnetic coupling for the three stacking configurations corresponding to local minima in total energy: AA stacking, where the Cr atoms of the top layer (orange) lie exactly on top of those of the bottom layer (blue); Monoclinic (M) stacking, where the Cr atoms of the top layer are shifted by [0,1/3] (in units of **a** and **b**) with respect to the bottom layer; AB stacking, where one of the Cr atoms of the top layer lies above the center of the hexagons in the bottom layer (i.e. with a relative shift by [1/3, 2/3]). DFT predicts that AB stacking favors ferromagnetic (FM) interlayer magnetic coupling, while AA and M stackings lead to antiferromagnetic (AFM) ordering. **c** Color plot of the total energy $E_{min}$ (the minimum energy between FM and AFM configurations, with zero set at the AB FM stacking), as a function of interlayer shift along the two lattice vectors, showing three non-equivalent local minima (indicated by AA, M and AB). **d** Total energy along the gray dashed line in panel (**c**) (the three non-equivalent local minima are indicated by red arrows). **e** Color plot of the effective interlayer exchange energy, $J_L = (E_{FM} - E_{AFM})/2$, as a function of interlayer shift. The orange regions correspond to AFM ($J_L > 0$) interlayer coupling while the blue regions correspond to FM ($J_L < 0$). **f** The interlayer exchange energy along the gray dashed line path in panel e.

occurring at the same values of $H$ and twice those values is a clear manifestation of spin-flip transitions, typical of AFM coupled layers with uniaxial magnetic anisotropy. The fact that two different field values are observed (0.2 T and 0.55 T) indicates the occurrence of two distinct types of AFM couplings in $CrBr_3$ devices. With reference to the magnitude of the field, we abbreviate them as L-type ("large") and S-type ("small") AFM coupling. The histogram in Fig. 3c gives clear statistical indications as to the occurrence of L and S jumps. S-type jumps at 0.2 T occur with nearly the same frequency as L-type jumps at 0.55 T. However, the number of jumps with "twice" the field (i.e., jumps at 0.4 T and 1.1 T) differs in the two cases. Only two out of seven devices that show a jump at 0.2 T also show a jump at 0.4 T, indicating that most commonly, only two layers are stacked in the way giving S-type coupling, and that longer sequences occur more rarely. On the contrary, most of the devices (6 out of 8) exhibiting a jump at 0.55 T also exhibit a jump at 1.1 T, indicating that for the stacking leading to L-type magnetoconductance jumps, sequences of four or more layers can be found relatively easily. These observations indicate that long sequences of $CrBr_3$ layers stacked in the way needed to produce S-type jumps are energetically more costly than for the other types of stacking (the stacking producing L-type magnetoconductance jumps and those giving rise to ferromagnetism), which is why they occur more rarely.

**Interlayer stacking of magnetic phases**

We attribute the occurrence of two distinct AFM phases to the presence in the tunnel barriers of layer sequences with two different layer stacking, resulting in different interlayer exchange couplings. To confirm the occurrence of different stackings in $CrBr_3$ devices with FM interlayer coupling, or L/S-type AFM interlayer coupling, we performed Raman spectroscopy at 10 K (Fig. 3d and Supplementary Fig. 3). Measurements with either parallel (XX configuration) or perpendicular (XY configuration) polarization of the incident and detected light were done, focusing on the modes in the 130–170 cm⁻¹ range[48,49], predicted to be particularly sensitive to the stacking (details are provided in the method section). To allow a direct comparison, the Raman data shown have been measured in all cases on four-layer tunnel barriers. The Raman spectra of a $CrBr_3$ FM device (magneto-conductance data shown in Fig. 2e, f) show two peaks at -142 cm⁻¹ and 152 cm⁻¹ that can be assigned to two twofold degenerate $E_g$ modes, whose position and intensity are independent of the polarization alignment (top panel, Fig. 3d). This is consistent with AB stacked $CrBr_3$, as already discussed in the literature[48,49]. In contrast, the sample in which L-type switching is observed (see Sample 10 in Supplementary Fig. 2 for detail) exhibits two additional peaks (-146 cm⁻¹ and 160 cm⁻¹) when measured in the XX configuration, whose relative intensity changes in the XY configuration (middle panel, Fig. 3d). This behavior is indicative of layers with monoclinic (M) stacking (Fig. 1b), whose broken rotational symmetry results in the splitting of the two twofold degenerate $E_g$ modes[50–53]. Finally, for the sample exhibiting S-type switching (see Sample 8 in Supplementary Fig. 2 for detail), again only two peaks are observed at positions close to (but not identical) to those of FM $CrBr_3$, independently of the polarization configuration employed, which is expected for AA stacking (Fig. 1b) with three-fold rotational symmetry.

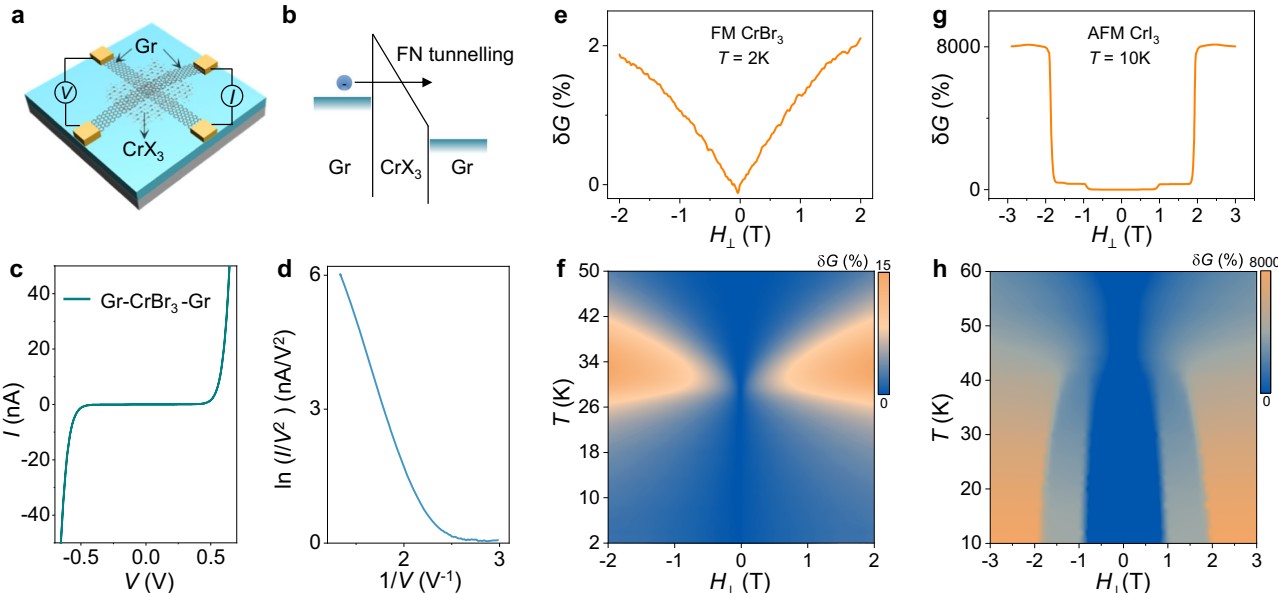

**Fig. 2 | Probing interlayer magnetic coupling of 2D magnets via tunneling magnetoconductance. a** Schematics of the tunnel junction devices, where electrons tunnel between two graphite sheets (Gr) separated by a magnetic Chromium trihalides ($CrX_3$) tunnel barrier. **b** Schematic energy diagram of the tunnel junctions illustrating the Fowler-Nordheim (FN) tunneling regime, with the electric field generated by the applied bias ($V$) that tilts the conduction band, causing the tunneling probability to increase exponentially. **c** Low-temperature ($T = 2$ K) tunneling current ($I$) across a four-layer $CrBr_3$ tunnel barrier as a function of applied voltage with $\ln(I/V^2)$ scaling linearly with $1/V$ for sufficiently large bias, as shown in (**d**). **e** magnetoconductance $\delta G(H, 2$ K) measured across the four-layer ferromagnetic

(FM) $CrBr_3$ tunnel barrier ($V = 0.6$ V), exhibiting only a small change (2%) as a function of magnetic field (in all measurements shown in this figure, the magnetic field ($H$) is applied perpendicular to the **a**, **b** plane of $CrBr_3$ ($H_\perp$)). **f** Color plot of the magnetoconductance $\delta G(H, T)$, showing the "lobes" around $T_C$ characteristic of ferromagnetic barriers. **g** $\delta G(H,10$ K) measured across an antiferromagnet (AFM) $CrI_3$ tunnel barrier (-7 nm, $V = 0.5$ V), showing two characteristic spin-flip transition fields (jumps) at 0.9 T and twice this value 1.8 T. **h** Color plot of $\delta G(H,T)$ for the same $CrI_3$ tunnel barrier, showing the evolution of the spin-flip transition fields with temperature.

The magnetotransport measurements and the observed Raman spectra are fully consistent with the density functional theory (DFT) calculations in ref. 24, which predict three local minima in the total energy of $CrBr_3$, corresponding to distinct stacking configurations, with the interlayer coupling that is FM for one and AFM for the other two (Fig. 1; the number of locally stable configurations is three also in other DFT studies[3,25], but the sign of the interlayer exchange differs, depending on details of the calculations). According to ref. 24, one of the two AFM stackings (AA, Fig. 1b) has a high symmetry (and should therefore give only two degenerate $E_g$ modes) and has a total energy that is sizably larger than the other two stacking (which is why long sequences of layers are found less frequently). Therefore, we attribute S-type magnetoconductance jumps (at 0.2 T and 0.4 T) to AA stacking, and the L-type jumps (at 0.55 T and 1.1 T) to monoclinic stacking. This attribution is consistent with the observed Raman spectra, because the monoclinic stacking has relatively lower symmetry (Fig. 1b) and gives rise to additional Raman peaks resulting from the splitting of the degenerate $E_g$ mode[50–53]. Note that the L-type jumps perfectly match the critical field previously observed in $CrBr_3$ bilayers synthesized by molecular beam epitaxy[45], although our analysis -in particularly the Raman data- attributes it to a different configuration with respect to that suggested in ref. 45. We conclude that the DFT calculations in ref. 24. capture all key aspects of the relation between structure and magnetism in $CrBr_3$ multilayers.

We emphasize that–despite the very systematic behavior of the jumps in magnetoconductance that only occur at four selected values of magnetic field, as expected for two distinct types of AFM multilayers –our observations pose some questions as to how regions exhibiting different magnetic orders are magnetically coupled to each other. If the regions are coupled either ferromagnetically or antiferromagnetically through one of the three stacking identified here

above, a simple analysis of the magnetic energy of multilayers containing multiple types of stacking indicates that magnetoconductance jumps should be expected to occur at many different magnetic fields determined by the precise stacking sequence of the entire multilayer, and not only at the fields observed in the experiments. Finding that jumps are only visible at the values expected for isolated multilayers of the different magnetic states seems to be only compatible with a scenario in which sequences with different stacking in a same multilayer are magnetically decoupled, so that they can re-orient independently when a magnetic field is applied. The decoupling probably originates from the presence of large misorientation (i.e., large twist angles) between layers that separate distinct stacking configurations, which occur spontaneously during the crystal growth process (i.e., whenever it occurs during the growth, such a misorientation between adjacent layers makes it energetically more favorable for the stacking sequence to change).

## Magnetic anisotropy and phase diagram

To further support the scenario outlined here above, we sought to realize tunnel barriers based on an isolated AFM multilayer, which allow us to test experimentally if indeed the magnetoconductance jumps are observed at the expected values in a device realized with a fully well-defined AFM structure. Given the relatively high probability to find long sequences of layers (4 L or thicker) with L-type stacking, we did succeed in realizing a perfectly L-type stacked 4 L tunnel barrier device (see also Supplementary Fig. 4) and in measuring its $T$- and $H$-dependent transport properties systematically. Fig. 4a–c compares the low-temperature ($T = 2$ K) magnetotransport measurements performed on this L-type 4 L barrier (green curves) with data measured on a "conventional" FM $CrBr_3$ 4 L tunnel barrier (orange curves). The 4L L-type barrier behaves precisely as anticipated, exhibiting

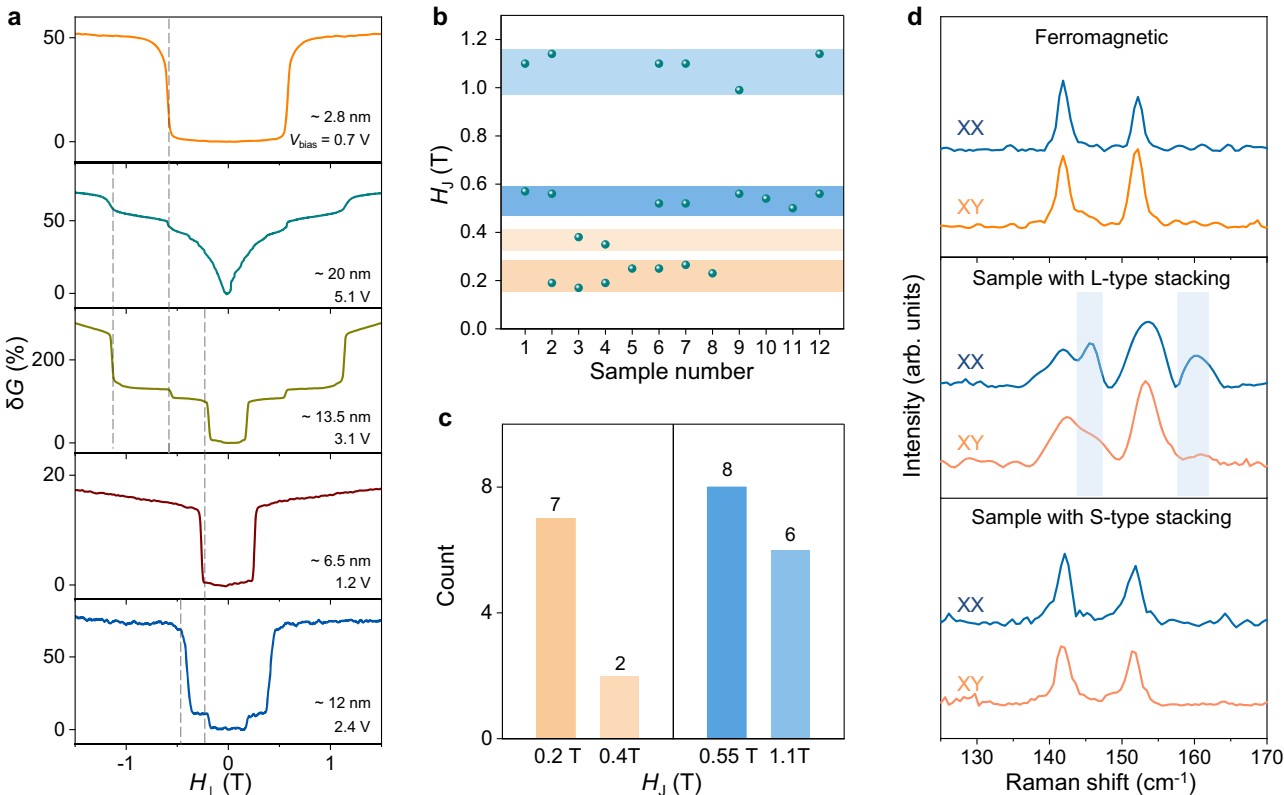

**Fig. 3 | Identification of CrBr₃ multilayers with different stacking order. a** Low-temperature tunneling magnetoconductance $\delta G(H_\perp)$ of five representative CrBr₃ multilayers, showing jumps in at several transition fields ($H_J$; indicated by the vertical gray dashed lines). **b** Summary of the different values of $H_J$ measured in twelve CrBr₃ multilayers (the magnetoconductance and thickness of each sample can be found in Supplementary Fig. 2). The values can be classified into two groups: 0.55 T and twice this value 1.1 T (blue rectangles), and 0.2 T and twice this value 0.4 T (orange rectangles). **c** Histogram of transition field distribution, as extracted from all devices measured. **d** Raman spectra of CrBr₃ multilayers with ferromagnetic

interlayer coupling (top panel), antiferromagnetic L-type ("large") stacking (middle panel) and antiferromagnetic S-type ("small") stacking (bottom panel), measured under both parallel (XX, blue lines) and crossed (XY, orange lines) polarization configurations at 10 K. The spectrum of the CrBr₃ layer with antiferromagnetic L-type stacking exhibits two additional peaks (indicated by the blue rectangles in the middle panel) compared to that of the multilayers with antiferromagnetic S-type stacking and ferromagnetic interlayer coupling, whose intensity varies in different polarized configurations, reflecting the lower symmetry of L- type stacking.

magnetoconductance jumps at 0.55 T and 1.1 T (i.e., the values identified above). The difference between the magnetoconductance of the AFM and FM barriers is obvious, irrespective of whether the magnetic field is applied perpendicular (Fig. 4a) or parallel (Fig. 4b) to the plane. For devices of the same thickness, the low temperature ($T = 2$ K) magnetoconductance is more than ten times larger for the AFM L-type device. When the field is applied in-plane, measurements show that the magnetoconductance exhibits no jumps and extends to a higher magnetic field (nearly 2 T), a consequence of the magnetic anisotropy in CrBr₃. Importantly, this observation indicates that the magnetic anisotropy in CrBr₃ is much smaller than in CrI₃, where the in-plane magnetoconductance extends up to 6 T[30], more than three times larger than in CrBr₃. Through a simple analysis that considers the joint effect of anisotropy and interlayer exchange, this difference implies that the magnitude of the uniaxial anisotropy in CrBr₃ is more than four times smaller than in CrI₃. Also, the temperature dependence of the tunneling conductance (Fig. 4c) is different for the FM and the L-type AFM barriers: in the FM barrier, the conductance increases when $T$ is lowered below $T_C$ (~31 K, smaller than that of thick layers) whereas for the AFM L-type barrier, a steeper decrease in conductance occurs below Néel temperatures $T_N^L$ (~29 K), again, similar trend as in CrI₃[30]. In short, the differences between L-type CrBr₃ and CrI₃ barriers are exclusively of quantitative nature, with CrI₃ exhibiting larger interlayer exchange, magnetic anisotropy, and magnetoconductance, trends that are all captured by the DFT-based calculations[24].

Full measurements of the conductance as a function of $T$ and $H$ are shown in Fig. 5, and exhibit all trends expected for a layered antiferromagnet. Whereas for $H = 0$ T, the tunneling conductance decreases below $T_N$ upon cooling, it increases when a sufficiently large magnetic field (-1 T) is applied (Fig. 5a). In addition, as $T$ is increased, the magnetoconductance becomes smaller and the jumps shift to lower magnetic field (Fig. 5b). The color plot of the magnetoconductance as a function of $T$ and $H$ in Fig. 5c—which represents the magnetic phase diagram of the L-type 4 L barrier—summarizes the results, and shows the phase boundaries separating the different magnetic states: the AFM phase at low field (I), the intermediate regime with the magnetization of the outer layers flipped (II) and the spin-flip phase at high field (III).

Having access to tunnel barriers made of different allotropes of CrBr₃, with different magnetic states, enables their critical temperatures to be compared. That is why—after determining $T_C$ for the FM structure and $T_N$ for the L-type AFM state—we measured the temperature and magnetic field dependence of the tunneling conductance in a barrier showing S-type magnetoconductance jumps. Because long sequences of S-type stacking are rare, we could only measure a tunnel barrier with S-stacked bilayers close to one of the contacts and the remaining layers stacked in the structure producing ferromagnetism (see Supplementary Fig. 5). The $T$- and $H$-dependence of the magnetoconductance (Fig. 5d) then exhibits concomitantly signatures of S-type antiferromagnetism (visible jump at

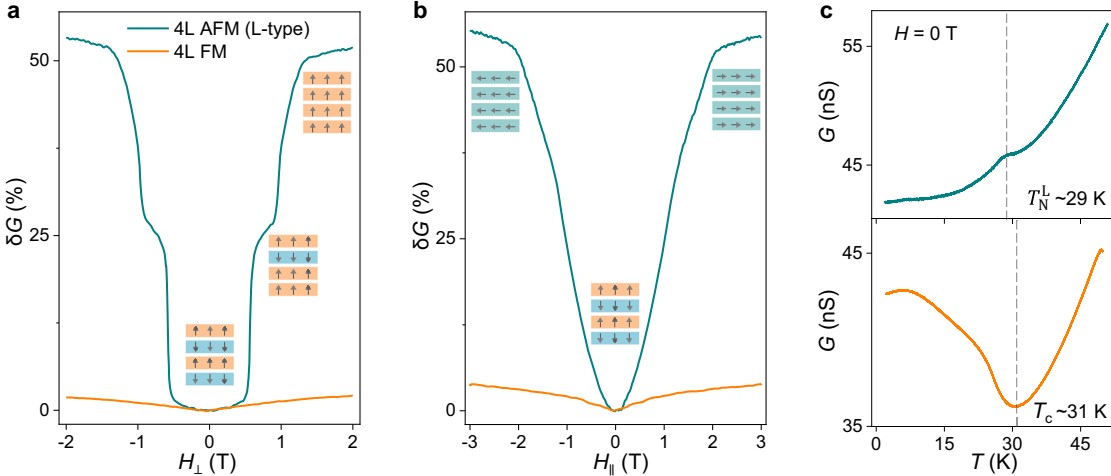

**Fig. 4 | Comparison of transport through four-layer CrBr₃ tunnel barriers with either ferromagnetic or L-type antiferromagnetic interlayer coupling. a** Out-of-plane ($H_\perp$) and **b** in-plane ($H_\parallel$) magnetic field dependence of the tunneling magnetoconductance ($\delta G(H, 2\,\mathrm{K}) = (G(H,T) - G_0(2\,\mathrm{K}))/G_0(2\,\mathrm{K})$) measured on the four-layer (4 L) CrBr₃ tunnel barriers with either L-type antiferromagnetic (AFM, green line) or ferromagnetic (FM, orange line) interlayer coupling. **c** Temperature

dependence of the tunneling conductance ($G$) at zero magnetic field (top panel: antiferromagnetic L-type stacking; bottom panel: ferromagnetic interlayer coupling). The measurements allow the determination of the Néel temperature ($T_N^L$ - 29 K; top panel) and Curie temperature ($T_C$ - 31 K; bottom panel), marked by the vertical dashed lines. In this and later figures, the bias voltage is 0.6 V and 0.7 V for the 4 L ferromagnetic and the L-type antiferromagnetic device, respectively.

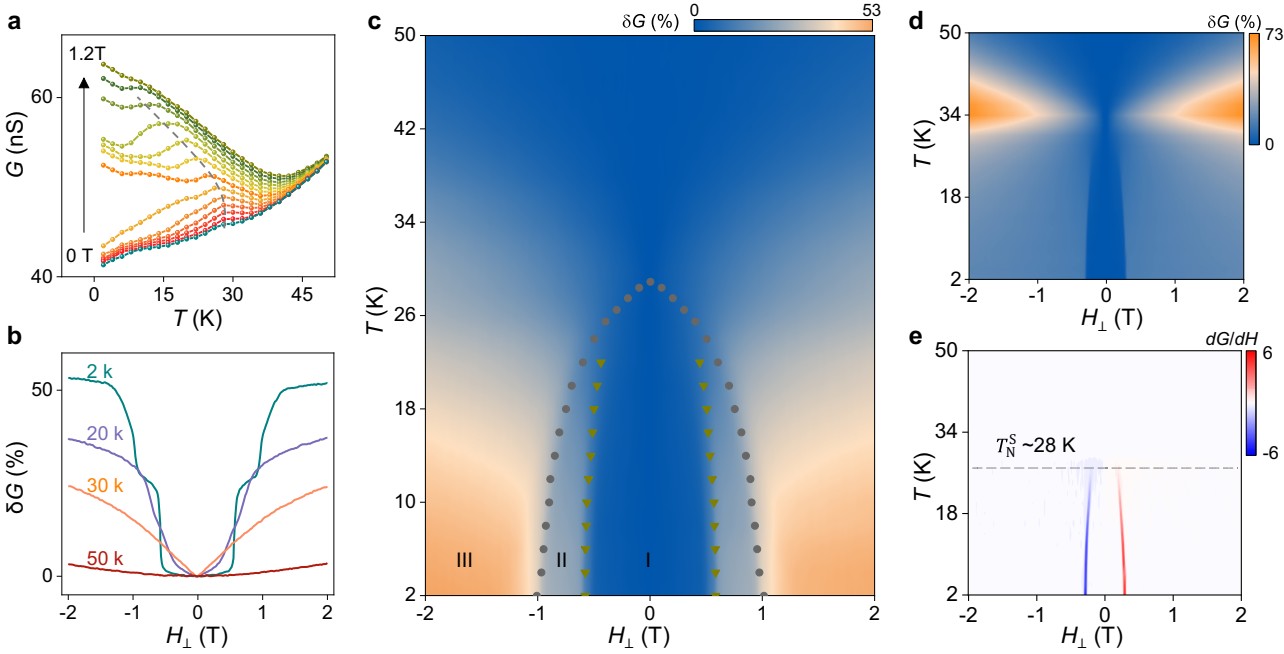

**Fig. 5 | Temperature dependent magnetotransport response of L-type and S-type antiferromagnetic CrBr₃ barriers. a** Temperature dependence of the tunneling conductance of a L-type antiferromagnetic four-layer CrBr₃ barrier measured at different magnetic fields (in all measurements shown in this figure, the magnetic field ($H$) is applied perpendicular to the **a**, **b** plane of CrBr₃ ($H_\perp$)). The kink in each curve (traced by the gray dashed line) follows the evolution of the onset of magnetic order with applied field. **b** Tunneling magnetoconductance ($\delta G$) plotted as a function of $H$ for selected values of $T$: as $T$ increases, $\delta G(H,T)$ decreases and the jumps shift to lower values of $H$. **c** Color plot of $\delta G(H,T)$, showing the phase diagram of four-layer antiferromagnetic L-type

stacked CrBr₃. The circles and triangles are extracted either from the $G$-vs-$T$ curves (see panel **a**) or from fields at which the magnetoconductance jumps occur (see panel **b**). **d** Color plots of $\delta G(H,T)$ and **e** $dG/dH(H,T)$ measured across on CrBr₃ multilayer tunnel barrier (~6.5 nm, $V = 1.2$ V) comprising layers with both ferromagnetic and S-type antiferromagnetic stacking (accounting for the concomitant presence of the characteristic magnetic lobes near the ferromagnetic $T_C$ and the jump characteristic of antiferromagnetic interlayer coupling). Tracking the temperature at which the magnetoconductance jumps disappear, we estimate the transition temperature ($T_N^S$) for S-type antiferromagnets to be ~28 K.

**Table 1 | Summary of CrBr$_3$ with different magnetic states**

| Stacking type | Symmetry information | Interlayer magnetic order | Estimated critical temperature | Magnetoconductance jumps |
|---|---|---|---|---|
| AA-stacking | $R\bar{3}$ | S-type AFM | ~ 28 K | ~ 0.2 T and 0.4 T |
| M-stacking | $C2/m$ | L-type AFM | ~ 29 K | ~ 0.55 T and 1.1 T |
| AB-stacking | $R\bar{3}$ | FM | ~ 31 K | No jumps |

$R\bar{3}$ and $C2/m$ are symbols of point group.

0.2 T at sufficiently low temperature) and of the FM state of CrBr$_3$ (the "lobes" of enhanced magnetoconductance near the Curie temperature at 32–34 K[40]). Although the concomitance of these different features makes a precise determination of the Néel temperature less straightforward, we find that the magnetoconductance jumps disappear at ~28 K (Fig. 5e). The Curie temperature of the FM state of CrBr$_3$ and the Néel temperatures of the L-type and S-type AFM states are therefore 31 K, 29 K and 28 K, values that agree with theory in multiple regards (we summarize the properties of the three magnetic states in Table 1). Specifically, the critical temperature values are all very close, because the critical temperature of weakly coupled 2D magnetic layers is primarily determined by intralayer interactions and depends only weakly on the interlayer interaction (irrespective of whether the interlayer coupling is FM or AFM). Indeed, the correction to the critical temperature of an isolated monolayer is predicted to scale with $|J_L/J|$ ($J_L$ and $J$ are the inter- and intra-layer exchange couplings)[54,55], a scaling consistent with our finding that S-type stacking, which exhibits a lower $T_N$ than L-type stacking, also exhibits a lower $J_L$ (as inferred from the smaller magnetic field at which the magneto conductance jumps occur). In addition, the Curie temperature of FM CrBr$_3$ is larger than Néel temperatures of both AFM states, in perfect agreement with DFT calculations that predict a sizably larger value of $J_L$ in the FM state than $|J_L|$ in the AA and M stacking configurations[24,25] (Fig. 1b).

## Discussion

Even though more direct experimental observations relating the different stackings of CrBr$_3$ multilayer tunnel barriers to their magnetoconductance properties would be desirable (e.g., by directly observing the layer stacking in the barriers used for transport measurements), our results provide a rather complete, and fully consistent characterization of stacking-dependent magnetism in CrBr$_3$ multilayers. Continuum models of small-angle and long-period moiré bilayers take the twist angle, the local strength of the interlayer exchange coupling as a function of layer registry, and the magnetic anisotropy as input, to calculate the expected magnetic state. For CrI$_3$ layers used to search for non-collinear magnetic phases in twisted bilayers, such a wealth of information is not available. For instance, the local strength of the interlayer exchange interaction as a function of relative shift between the layer is not known experimentally, and only two of the three expected magnetic states have been observed in experiments[29,56]. In addition, the large uniaxial magnetic anisotropy of CrI$_3$ reduces the portion of the phase diagram where the non-collinear phase can emerge, which imposes more stringent conditions on the twist angle. The work presented here clearly shows that the situation for CrBr$_3$ is different. Both the experimental observation of all three predicted locally stable magnetic states, and the overall agreement found with the calculations indicate that our understanding of interlayer exchange as a function of layer registry is in fact rather detailed and complete. The magnetic anisotropy of CrBr$_3$, being more than four times smaller than in CrI$_3$, is ideal and increases the parameter regime in which non-collinear magnetic phases can be found. We therefore conclude that CrBr$_3$ offers the most favorable conditions among all Chromium trihalides to controllably engineer and model non-collinear magnetic states in twisted bilayer structures.

## Methods

### DFT calculation

Density-functional-theory simulations are performed using the Quantum ESPRESSO distribution[57,58]. Van der Waals interactions between the layers are included through the spin-polarised extension[59] of the revised vdw-DF2 exchange-correlation functional[60,61], with a cutoff to truncate spurious interactions between artificial periodic replicas along the vertical direction[62–64]. A $8 \times 8 \times 1$ Γ-centered Monkhorst-Pack grid is adopted to sample the Brillouin zone. Pseudopotentials are chosen from the Standard Solid-State Pseudopotential (SSSP) accuracy library (v1.0)[65–67] with increased cutoffs of 60 Ry for wave functions and 480 Ry for density. In total energy calculations as a function of the relative displacement between the layers, intralayer atomic positions are kept fixed by considering the structure of DFT-relaxed monolayers with the experimental lattice parameter and interlayer separation. For refined results along high symmetry lines, atomic positions are relaxed until the force acting on each atom falls below a threshold of 26 meV/Å, while keeping fixed the in-plane coordinates of Cr atoms. The AiiDA materials informatics infrastructure[68,69] is adopted to manage and automate all calculations.

### Bulk crystal growth

CrBr$_3$ bulk crystals were grown by the chemical vapor transport method[70]. The elemental precursors Chromium (99.95% CERAC) and TeBr$_4$ (99.9% Alfa Aesar) were mixed with a molar ratio 1:0.75 to a total mass of 0.5 g, and were placed in a quartz tube with a length of 13 cm to achieve a temperature gradient of ~10 °C/cm from the hot end at 700 °C to the cold end at 580 °C. After seven days at this temperature, the furnace was switched off. When the tube cooled to room temperature, CrBr$_3$ crystals were found to crystallize towards the cold end of the tube, over a length that corresponded to a growth temperature range of ~650 °C–580 °C. As shown in Supplementary Fig. 1, Raman spectroscopy measurements show that different crystals harvested from a same batch can exhibit the coexistence of at least two different structures.

### Device fabrication

CrX$_3$ multilayers were first mechanically exfoliated from the bulk crystals, and tunnel junctions of multilayer HBN/graphene/CrX$_3$/graphene/HBN were assembled using a pick-and-lift technique[71] with stamps of PDMS/PC in a glove box filled with nitrogen gas. The thickness of CrX$_3$ multilayers was obtained by atomic force microscope measurements performed on the encapsulated devices. Edge contacts to the graphene multilayers were made by electron beam lithography, reactive-ion etching, electron-beam evaporation (10 nm Cr/50 nm Ar), and lift-off process. Transport measurements were performed in a homemade low-noise electronics system combined with a helium cryostat from Oxford Instruments.

### Raman measurement

All Raman spectroscopy measurements were performed using a Horiba system (Labram HR evolution) combined with a helium flow cryostat. The laser (532 nm, ~1 μm) was linearly polarized with its polarization angle controlled via a half-wave plate (Thorlabs) and was focused on the sample (inside the cryostat) through a 50× Olympus objective. The scattering light of the sample was collected by the same objective and passed through the analyzer, then was sent to a Czerni–Turner spectrometer equipped with a 1800 groves mm$^{-1}$

grating and was detected by a liquid nitrogen-cooled CCD-array. Measurements under either parallel (XX) or crossed (XY) polarization were performed by varying the half-wave plate while keeping the analyzer on the detecting light path fixed. Similarly to previous reports[50–53], the Raman tensors of doubly degenerate $E_{g1}$ and $E_{g2}$ modes (in the AB and AA stacking, $R\bar{3}$ group) and the non-degenerate $A_g$ and $B_g$ modes (in the monoclinic stacking, $C2/m$ group) can be derived as:

$$E_{g1} = \begin{pmatrix} m & n & p \\ n & -m & q \\ p & q & 0 \end{pmatrix}, E_{g2} = \begin{pmatrix} n & -m & -q \\ -m & -n & p \\ -q & p & 0 \end{pmatrix}, A_g = \begin{pmatrix} a & 0 & d \\ 0 & c & 0 \\ d & 0 & b \end{pmatrix}, B_g = \begin{pmatrix} 0 & e & 0 \\ e & 0 & f \\ 0 & f & 0 \end{pmatrix}$$

As a result, for AB and AA stacked multilayers, the Raman intensity for the $E_{g1}$ and $E_{g2}$ modes as a function of $\theta$ can be derived as: $I_{(Eg1)} \propto |m\sin(\theta) - n\cos(\theta)|^2$ and $I_{(Eg2)} \propto |m\cos(2\theta) + n\sin(2\theta)|^2$, where $\theta$ is the polarized direction of excitation light with respect to the analyzer. Thus, the dependence on the polarization angle cancels out when the two modes ($E_{g1}$ and $E_{g2}$) are degenerate, leading to one single $E_g$ peak (the total intensity of the degenerate modes is the same under either XX configuration or XY configuration; observed in Fig. 3d, top panel and bottom panel). However, for the monoclinic stacking, the degenerate $E_g$ modes split into the non-degenerate $A_g$ and $B_g$ modes. Since the $B_g$ mode is distinct from the $E_g$ mode and its Raman intensity as a function of $\theta$ can be expressed as: $I_{(Bg)} \propto e^2\cos^2(\theta)$, different intensities of the Raman peaks under the XX configuration and XY configuration are observed (Fig. 3d, middle panel).

## Data availability

The data supporting the findings of this study are available free of charges from the Yareta repository of the University of Geneva. (https://doi.org/10.26037/yareta:ydzxc5zwnfdv3p64o5zxtqb2vy).

## Code availability

All codes adopted for DFT calculation are open source and available at https://gitlab.com/QEF/q-e.

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

## Acknowledgements

The authors gratefully acknowledge Alexandre Ferreira for technical support. A.F.M. gratefully acknowledges the Swiss National Science Foundation (Division II, project #200020_178891) and the EU Graphene Flagship project for support. M.G. acknowledges support from the Italian Ministry for University and Research through the Levi-Montalcini program and through the PNRR project ECS_00000033_ECOSISTER. Z.W. acknowledges support from the National Natural Science Foundation of China (92265103), Shaanxi Fundamental Science Research Project for Mathematics and Physics (22JSYO26) and the Fundamental Research Funds for the Central Universities.

## Author contributions

F.Y. and Z.W. fabricated the devices and performed the transport measurements with the help of I.G.L. A.F.M. supervised the project. V.D. and F.Y. performed optical measurements with help of N.U. and J.T.; E.G. and F.W. grew the crystals. M.G. performed ab-initio calculations. A.F.M., F.Y., M.G., and I.G.L. analyzed the data and wrote the manuscript with input from all authors. All authors discussed the results.

## Competing interests

The authors declare no competing interests.
