## [Peer Review File · Nature Communications]

Reviewers' Comments:

Reviewer #2:

Remarks to the Author:

This paper presents interesting results on antiferromagnetic order in CrBr₃, which differs from the ferromagnetic phase of bulk CrBr₃. Samples grown by CVT and collected from different parts of the reactor show differing properties as measured by magnetoconductance measurements, where the CrBr₃ serves as a magnetic tunnel barrier between two graphene electrodes. Ferromagnetic samples show very little variation of conductance, while antiferromagnetic samples show large switching as the individual magnetic layer change their orientation. Out of 20 samples, 8 are ferromagnetic and the other 12 are categorized into two groups with large switching field ("L-type") or small switching field ("S-type"). Some samples show one switch and others show two switches (one even shows three). But across many samples, the field values for the switches are fairly consistent. Therefore this categorization of three types of samples, one ferromagnet and two distinct types of antiferromagnet, seems pretty reliable. In particular, one of these antiferromagnetic states (S-type) has not been observed before. The observation of all three magnetic states predicted is the main claim of their work. The observation of three distinct states is fairly well supported by the data. Their assignment to particular stacking sequences is less reliable.

Theoretical studies predict that AB stacking is ferromagnetic and monoclinic (M) and AA stacking are antiferromagnetic. The authors attribute the L-type behavior to the M (monoclinic) stacking and the S-type behavior to the AA stacking, based on Raman spectroscopy measurements. This seems to disagree with previous spin-polarized STM studies of CrBr₃, which attributes the L-type behavior to AA stacking. While the assignment of This brings me to my main questions.

1. How are the bulk crystals characterized? Have TEM been performed to justify the claim that some of the collected samples from CVT have a large density of stacking faults? Or is it simply an inference based on the idea that samples with antiferromagnetic order must have different stacking sequences? A direct experimental connection between the structure and magnetic properties would be good to have. This would not require necessarily performing TEM on the device under study, but even a TEM from the crystal that was used to make the device.

2. How should I picture the stacking sequence within a sample? For example, does the L-type sample consist of M stacking throughout, or is it mostly AB stacking (ferromagnetic) with M stacking every once in a while? If it is the latter, then I have questions about the relative consistency of the switching field. If I have two ferromagnetic layers coupled by an interfacial antiferromagnetic coupling, I would expect the switching field to decrease with the ferromagnetic layer thickness because the Zeeman energy increases with ferromagnetic layer thickness, while the antiferromagnetic coupling is independent of the ferromagnetic layer thickness. The relative consistency of the switching field across samples would suggest that the spacing of M stackings (where most stackings are AB) is relatively the same in all samples. Is that a correct way to interpret this?

3. The S-type sample Raman data looks very similar to the AB stacking, while the L-type sample Raman looks very different. What are the thicknesses of the samples? Is the S-type sample expected to the AA stacking throughout? Is the L-type sample expected to the M stacking throughout? These are, of course, closely related to my point #2 above. I do understand the argument that the M symmetry is very different from the AA and AB, so I think the rationale for the assignment of L-type to M and S-type to AA is reasonable. But the lack of direct structural characterization (i.e. TEM) leaves room for questions.

Lastly, there is a lot of discussion of weaker magnetic anisotropy compared to CrI₃ and how the results in this paper are relevant for achieving non-collinear magnetic phases. I understand the need nowadays to emphasize the connection to bigger potential results in the future, but this might also have the effect of painting the current work as a stepping stone. Looking beyond such claims, I do think the current work is interesting on its own and Nature Communications could be an appropriate venue. A better discussion of the structure and possibly some additional supporting

data would improve the quality of the work.

Reviewer #3:

Remarks to the Author:

The manuscript is well-written and presents a rather thorough and elegant examination of the magnetic properties of exfoliated CrBr₃ multilayers using magnetoconductance measurements and Raman spectroscopy. Three different stacking patterns are found, consistent with DFT calculations. The authors also identify that the uniaxial anisotropy in CrBr₃ is much smaller than that in CrI₃, making CrBr₃ a favorable platform to examine non-collinear magnetic orderings in its potential moire heterostructures. The findings are both interesting and timely, making this manuscript a valuable contribution to the field. I recommend this manuscript for publication with only some optional changes.

Firstly, to help readers quickly grasp the key features of each stacking pattern, it could be beneficial to include a summary of the properties of the three different stacking patterns in a table. Alternatively, the authors could add additional information in Figure 1b, such as the symmetry information, critical fields, Neel/Curie temperatures, etc., near the corresponding panels.

Secondly, the authors could consider including a discussion on the potential use of magnetoconductance in studying other van der Waals magnets. This would further emphasize the broader implications of their findings and make the manuscript more relevant to a wider audience.

Overall, the manuscript is of high quality, and these optional changes will only serve to enhance its readability and impact.

Reviewer #4:

Remarks to the Author:

The manuscript describes temperature dependent TMR data of 20 G/CrBr₃/G devices compared with Raman data and DFT calculations used from a previous publication. It shows convincingly that three different stackings are related to three different interlayer couplings in very good agreement with the DFT predictions. These results make CrBr₃ a favorable system for designing noncollinear magnetism in twisted bilayers. This will definitely motivate a number of additional future studies using the material.

The quality of the data is excellent and convincing, very nicely supporting the claims. The manuscript is well written and largely gives all details that are necessary to reproduce the experiments.

I would fully support publication in Nature Communications provided that the comments below have improved the manuscript adequately.

- 1) Throughout the manuscript, it is not clear if the stacking faults are only deduced from the MR and Raman data or if any additional evidence for stacking faults in the CrBr₃ used by the authors exists. A paper from Manchester is quoted that evidences stacking faults by TEM, but I guess the authors do not use the same batch of materials. Moreover, it is speculated that different temperatures in the quartz tube during flux growth are decisive for the number of stacking faults, but it is not clear, if the used material is from a certain position of the quartz tube. Please describe more carefully throughout the manuscript, what is known about stacking faults in the particular batch of material.

One example: page 6, line 130: „ferromagnetic multilayers, with no stacking faults“

Is this deduced from the ferromagnetic behaviour without steps in MR or is there any other indication that the sample has no stacking faults ?

- 2) Often, the voltage used for the plotted MR or T -dependent resistance data is not given (Fig. 2e-h, Fig. 3a, Fig. 4a-c, Fig. 5, Fig. S1). Please add and, if available comment on the voltage dependence.
- 3) The spot size of the Raman experiments would be helpful and, if it is smaller than the devices, a comment on the homogeneity of the Raman spectra would be adequate.
- 4) There are a few positions, where quotations should be added to substantiate the statement:
 - a) Page 5, line 103: formula for Fowler-Nordheim tunneling
 - b) Page 5, line 106: „magnetic state determines height of the tunnel barrier“
 - c) Page 5, line 113: „lobes“ ... appear .. due to the divergence of magnetic susceptibility
 - d) page 10, line 262: two of the three expected magnetic states have been observed

Minor:

- 1) It would be nice to add the thicknesses of the CrBr₃ samples directly to Fig. 3b, e.g. vertically on top. This enables easy direct comparison.
- 2) Absolute numbers of δG in Fig. 3a would be better. There is no reason to hide this information in the supplement.
- 3) The label „8k“ in Fig 1g/h is irritating. At least, in the caption, $8k=8000$ should be mentioned.
- 4) I would prefer that all x-axis with out-of-plane H are labelled H_{\perp} as in Fig. 4a. This would make things less ambiguous.
- 5) Labeling Fig. 1b with AFM(L-type) and AFM(S-type) would make the reading of the paper easier. Else, one has to continuously translate between structural and magnetic labels.
- 6) Two typos:
 - a) page 3, line: 53: ferro and antiferro...
 - b) page 12, line 299: 10·C, 700·C, 580·C

Reply to Referee #2

Comment 1: *This paper presents interesting results on antiferromagnetic order in CrBr₃, which differs from the ferromagnetic phase of bulk CrBr₃. Samples grown by CVT and collected from different parts of the reactor show differing properties as measured by magnetoconductance measurements, where the CrBr₃ serves as a magnetic tunnel barrier between two graphene electrodes. Ferromagnetic samples show very little variation of conductance, while antiferromagnetic samples show large switching as the individual magnetic layer change their orientation. Out of 20 samples, 8 are ferromagnetic and the other 12 are categorized into two groups with large switching field (“L-type”) or small switching field (“S-type”). Some samples show one switch and others show two switches (one even shows three). But across many samples, the field values for the switches are fairly consistent. Therefore this categorization of three types of samples, one ferromagnet and two distinct types of antiferromagnet, seems pretty reliable. One of these antiferromagnetic states (S-type) has not been observed before. The observation of all three magnetic states predicted is the main claim of their work. The observation of three distinct states is fairly well supported by the data. Their assignment to particular stacking sequences is less reliable.*

Reply 1: We are delighted to hear that the referee found our results interesting. We thank the reviewer for correctly summarizing the main point of the paper, namely that three magnetic states are found in the exfoliated CrBr₃ multilayers. The reviewer also has some concerns and questions, which we address point-by-point in the following.

Comment 2: *Theoretical studies predict that AB stacking is ferromagnetic and monoclinic (M) and AA stacking are antiferromagnetic. The authors attribute the L-type behavior to the M (monoclinic) stacking and the S-type behavior to the AA stacking, based on Raman spectroscopy measurements. This seems to disagree with previous spin-polarized STM studies of CrBr₃, which attributes the L-type behavior to AA stacking. While the assignment of this brings me to my main questions.*

Reply 2: The referee correctly mentions—as we also wrote in the original manuscript—that earlier STM studies assign L-type behaviour to a layer stacking (neither AA nor M stacking) different from ours. We find that the attribution in the STM study is questionable, because the same study assigns the ferromagnetic phase of CrBr₃ to a stacking different from the one that is well established from the study of the bulk structure (AB stacking). Indeed, an unambiguous attribution of structure based on STM is always difficult, even on materials much simpler than CrBr₃.

Note also that another study has been reported based on transmission electron microscopy (TEM; J. Am. Chem. Soc. 2023, 145, 3624–3635) on exfoliated layers, in which unambiguous bilayer stacking configurations matching the ones we claim in our manuscript (AB stacking and monoclinic stacking) have been observed. The AA stacking configuration, instead, was not

observed. This is also consistent with the conclusions of our work, namely that the much higher total energy of AA stacking as compared to the other two stacking configurations makes AA stacking sequences much more difficult to observe (indeed, in our observations, long sequences of CrBr₃ layers stacked to give the S-type configuration, resulting in a jump around 0.4 T in magnetoconductance, were also rarely observed).

Comment 3: *1. How are the bulk crystals characterized? Have TEM been performed to justify the claim that some of the collected samples from CVT have a large density of stacking faults? Or is it simply an inference based on the idea that samples with antiferromagnetic order must have different stacking sequences? A direct experimental connection between the structure and magnetic properties would be good to have. This would not require necessarily performing TEM on the device under study, but even a TEM from the crystal that was used to make the device.*

Reply 3: We have used Raman spectroscopy to analyse multiple crystals from multiple growth batches. What we find is that some crystals only show peaks at the positions expected for the thermodynamically stable ferromagnetic phase (see Fig.R1 (a)), while other crystals exhibit a clearly visible additional peak at ~161 cm⁻¹(see Fig.R1 (b)), i.e., at almost the same position of one peak that we detect in L-type CrBr₃. We attribute the additional Raman peak that –when present –is always located at the same energy to a secondary material phase with different structure, whose amount is sufficiently large to give a clearly detectable signal. That is what we meant when we wrote that we select crystals with higher density of stacking faults to realize tunnel barriers. We now make more precise statements in the main text to avoid misunderstandings, and show Raman data from bulk crystals with and without the peak at 161 cm⁻¹ in the supplementary information.

Let us further mention that comparable Raman signal was observed in crystals that we grew in Geneva and in crystals that we purchased from HQ Graphene, suggesting that the secondary phase of CrBr₃ can be quite commonly found. Although we ourselves are not equipped to perform TEM measurements, we emphasize that the conclusion that we draw from Raman spectroscopy is fully in line with that drawn from TEM experiments reported in the literature, already mentioned above (J. Am. Chem. Soc. 2023, 145, 3624–3635; Nano Lett 2020,20, 6582).

Finally, we believe that performing TEM experiments on the same multilayers employed in the actual tunnel barriers used for transport measurements –which is what ideally would be desirable to directly connect transport measurements and magnetic structure– is prohibitively difficult. That is because CrBr₃ is very sensitive to ambient exposure and cutting a slice of a device to enable TEM measurements without causing significant CrBr₃ degradation seems extremely challenging.

We have modified the manuscript in several points to make clear what we meant when we state that barriers have been realized using multilayers exfoliated from bulk crystals containing

different amounts of different structural phases. We also make more explicit the statement that attributing the different Raman signals to different structures is an assumption. Finally, in the last paragraph, we mention that despite the full consistency of our results, direct experimental observations linking structure and magnetoconductance would be desirable.

Fig. R1. Raman spectroscopy performed at different positions of two representative bulk crystals at $T = 10$ K. (a) Raman spectroscopy measurements performed on different positions of a same crystal show exclusively the peaks expected from the thermodynamically stable (ferromagnetic) structural phase. (b) Crystals exhibiting an additional Raman peak at approximately 161 cm^{-1} (indicated by blue rectangle) can be found in all growth batches, as well as in crystals purchased from commercial providers. We attribute this additional peak to the presence of a considerable amount of a second allotrope of CrBr_3 with a different layer stacking.

Comment 4: 2. How should I picture the stacking sequence within a sample? For example, does the L-type sample consist of M stacking throughout, or is it mostly AB stacking (ferromagnetic) with M stacking every once in a while? If it is the latter, then I have questions about the relative consistency of the switching field. If I have two ferromagnetic layers coupled by an interfacial antiferromagnetic coupling, I would expect the switching field to decrease with the ferromagnetic layer thickness because the Zeeman energy increases with ferromagnetic layer thickness, while the antiferromagnetic coupling is independent of the ferromagnetic layer thickness. The relative consistency of the switching field across samples would suggest that the spacing of M stackings (where most stackings are AB) is relatively the same in all samples. Is that a correct way to interpret this?

Reply 4: In the original manuscript we had not commented on the nature of the magnetic coupling between the different parts of a same multilayer that have different stacking and different magnetic order. However, we agree with the referee that this is a relevant issue that we should have commented about.

What the referee writes is entirely correct: if the differently magnetically ordered part of a same multilayer are magnetically coupled, the switching fields can be determined by a straightforward calculation to balance Zeeman and exchange energy in configurations where the magnetization of every layer can point up or down. It is simple to show that –for a generic multilayer that combines at least two different magnetic structures (such as the ones that we discuss in our manuscript) –such a calculation would then lead to switching fields that can take virtually any value (especially if the multilayer is thick), depending on the precise thickness and stacking sequence. This, however, is clearly not what we see in experiments, where the magnetoconductance jumps observe in all devices occur at only 4 distinct values that are always the same.

The only way in which we can reconcile the magnetic energy calculations with the experimental observations is by assuming that the parts of a same multilayer that have different magnetic order (and hence different stacking) are decoupled magnetically, so that when a magnetic field is applied switch independently. We believe that this is indeed what happens. The scenario that we envision is that during the growth a certain stacking sequence stops when a layer forms that is rotated by a large angle (relatively to the uppermost layer already deposited), and that the material continues to grow with a different layer stacking (until a new layer again grows with a large twist angle). A large twist angle not only decouples the structure but also strongly suppresses the magnetic coupling between the adjacent layers (i.e., the effective interlayer exchange coupling), since in the presence of a large twist angle the atoms are not any more in registry. As a result, different parts of the multilayers are magnetically decoupled and switch independently one of the other when a magnetic field is applied. A direct experimental validation of this scenario would be to perform TEM on the same multilayers used to realize tunnel barrier devices, but as we explained above this is prohibitively difficult owing to the air sensitivity of CrBr₃ (and irrespective of feasibility, we are not equipped to do such TEM experiment).

In the revised version of the manuscript, we have added an entire new paragraph to address the issue of magnetic coupling between different parts of a same CrBr₃ multilayer, to explain what we just wrote here above.

Comment 5: 3. *The S-type sample Raman data looks very similar to the AB stacking, while the L-type sample Raman looks very different. What are the thicknesses of the samples? Is the S-type sample expected to the AA stacking throughout? Is the L-type sample expected to the M stacking throughout? These are, of course, closely related to my point #2 above. I do understand the argument that the M symmetry is very different from the AA and AB, so I think the rationale for the assignment of L-type to M and S-type to AA is reasonable. But the lack of direct structural characterization (i.e. TEM) leaves room for questions.*

Reply 5: Both the L-type and S-type samples whose Raman data are shown in Fig. 3 have a thickness of 2.8 nm (four layers). They both also do not have complete M- or AA- stacking, as they both show only one jump in magnetoconductance. For these reasons, it should be expected that Raman experiments on these two devices have comparable sensitivity to the two different stacking expected on the basis of their magnetoconductance.

We do agree that a direct structural characterization –if possible –would provide stronger evidence than Raman measurements, but as we mentioned already above, that is not within our capabilities. Let us also emphasize that our assignment of the structure to the different magnetic state is not based exclusively on Raman, but also on the full consistency of our analysis. An important point is that long sequences of one of the two antiferromagnetic structures are found more rarely, which is consistent with the higher formation energy of the structure given by ab-initio calculations. Another consistent aspect is the slightly lower critical temperature that we observe, which is consistent with the smaller interlayer exchange coupling.

We have now emphasized in the conclusions that what we present is a fully consistent scenario that explains our observations and that if possible, it would be highly desirable in the future to determine directly the relation between structure and magnetic properties by correlating a direct experimental observation of the individual different structures with their magnetic properties.

Comment 6: *Lastly, there is a lot of discussion of weaker magnetic anisotropy compared to CrI3 and how the results in this paper are relevant for achieving non-collinear magnetic phases. I understand the need nowadays to emphasize the connection to bigger potential results in the future, but this might also have the effect of painting the current work as a stepping stone. Looking beyond such claims, I do think the current work is interesting on its own and Nature Communications could be an appropriate venue. A better discussion of the structure and possibly some additional supporting data would improve the quality of the work.*

Reply 6: We fully agree with the referee that the results that we present are interesting on their own right. That is why we spent a long time carrying out these experiments, making 20 different devices to gather enough statistics on the different magnetoconductance response of different CrBr₃ tunnel barriers, characterizing the structures to the best of our abilities, and carrying out all the needed ab-initio calculations. We are glad to read in the comment of this referee –as well as of the others– that the relevance and interest of our results come across clearly from our manuscript. That said, it is also the case –as we emphasize in the introduction– that our work shows how CrBr₃ is the ideal system to search for highly interesting, recently predicted non-collinear magnetic phases. We think that it is important to emphasize that aspect very explicitly as well, because making this clear may motivate other groups elsewhere to exploit CrBr₃ to realize these non-collinear phases experimentally.

Reply to Referee #3

Comment 1: *The manuscript is well-written and presents a rather thorough and elegant examination of the magnetic properties of exfoliated CrBr₃ multilayers using magnetoconductance measurements and Raman spectroscopy. Three different stacking patterns are found, consistent with DFT calculations. The authors also identify that the uniaxial anisotropy in CrBr₃ is much smaller than that in CrI₃, making CrBr₃ a favorable platform to examine non-collinear magnetic orderings in its potential moire heterostructures. The findings are both interesting and timely, making this manuscript a valuable contribution to the field. I recommend this manuscript for publication with only some optional changes.*

Reply 1: We thank the referee for the positive comments on our work and for their recommendation to publish our manuscript.

Comment 2: *Firstly, to help readers quickly grasp the key features of each stacking pattern, it could be beneficial to include a summary of the properties of the three different stacking patterns in a table. Alternatively, the authors could add additional information in Figure 1b, such as the symmetry information, critical fields, Neel/Curie temperatures, etc., near the corresponding panels.*

Reply 2: We thank the referee for this suggestion. To help readers grasp the key features of each stacking pattern, we have added the tables as shown below in the revised manuscript.

Table 1 Summary of CrBr ₃ with different magnetic states				
Stacking type	Interlayer magnetic order	Symmetry information	Estimated critical temperature	Magnetoconductance jumps
AA-stacking 	S-type AFM 	$R\bar{3}$	~ 28 K	~ 0.2 T and 0.4 T
M-stacking 	L-type AFM 	$C2/m$	~ 29 K	~ 0.55 T and 1.1 T
AB-stacking 	FM 	$R\bar{3}$	~ 31 K	No jumps

Comment 3: *Secondly, the authors could consider including a discussion on the potential use of magnetoconductance in studying other van der Waals magnets. This would further emphasize the broader implications of their findings and make the manuscript more relevant to a wider audience.*

Reply 3: We thank the referee for this suggestion. Tunnelling magnetoconductance measurements have been widely used in probing the magnetic properties of atomically thin

magnetic insulators. It has been shown that magnetic phase boundaries (and even the complete magnetic phase diagram) of insulating atomically thin magnets can be detected by using them as tunnel barriers and measuring their temperature-dependent magnetoconductance in previous literatures (Nat Commun, 2018, 9, 2516; Science, 2018, 360, 1214; Science, 2018, 360, 1218; Nat Nanotechnol, 2019, 14, 1116; Nat Commun 2021, 12, 6659; Adv. Mater., 2022, 34, 2204940; Nano Lett. 2022, 22, 15, 6149 and doi:10.1002/adma.202211653). Therefore, considering that this topic has been thoroughly discussed and it is not a specific novelty of the manuscript under consideration, we prefer not to repeat this information, also to limit the length of the manuscript.

Comment 4: *Overall, the manuscript is of high quality, and these optional changes will only serve to enhance its readability and impact.*

Reply 4: We appreciate the referee's time and efforts in reviewing our manuscript and especially the valuable suggestion that help us to improve our manuscript.

Reply to Referee #4

Comment 1: *The manuscript describes temperature dependent TMR data of 20 G/CrBr₃/G devices compared with Raman data and DFT calculations used from a previous publication. It shows convincingly that three different stackings are related to three different interlayer couplings in very good agreement with the DFT predictions. These results make CrBr₃ a favorable system for designing noncollinear magnetism in twisted bilayers. This will definitely motivate a number of additional future studies using the material. The quality of the data is excellent and convincing, very nicely supporting the claims. The manuscript is well written and largely gives all details that are necessary to reproduce the experiments. I would fully support publication in Nature Communications provided that the comments below have improved the manuscript adequately.*

Reply 1: We thank the referee for the positive comments on our work and recommendation for its publication after further revision.

Comment 2: *1) Throughout the manuscript, it is not clear if the stacking faults are only deduced from the MR and Raman data or if any additional evidence for stacking faults in the CrBr₃ used by the authors exists. A paper from Manchester is quoted that evidences stacking faults by TEM, but I guess the authors do not use the same batch of materials. Moreover, it is speculated that different temperatures in the quartz tube during flux growth are decisive for the number of stacking faults, but it is not clear, if the used material is from a certain position of the quartz tube. Please describe more carefully throughout the manuscript, what is known about stacking faults in the particular batch of material.*

One example: page 6, line 130: „ferromagnetic multilayers, with no stacking faults“ Is this deduced from the ferromagnetic behaviour without steps in MR or is there any other indication that the sample has no stacking faults ?

Reply 2: We agree that the phrasing used in the originally submitted version of our manuscript was not sufficiently clear and ambiguous as to what is factually known about the presence of different structural phases in our CrBr₃ crystals. We have modified the phrasing in multiple parts, to make more factual statements, free of ambiguity. We have also added data to the supplementary information, to show Raman spectra on different bulk crystals.

Specifically, we find that different bulk crystals show two different types of Raman spectra. Some crystals only exhibit peaks consistent with the known thermodynamically stable structure of the material (i.e., the structure exhibit ferromagnetism; see Fig. R2(a)). Other crystals show an additional peak at 161 cm⁻¹ (see Fig.R2 (b)), which essentially coincide with one of the peaks that we measure on L-type multilayers. We attribute this additional Raman peak to a different allotrope of CrBr₃, present in sufficient quantity to give a clearly measurable signal in bulk crystals. That is how we can select crystal for exfoliation to maximize the probability to find non-ferromagnetic CrBr₃ structures. This conclusion is fully consistent with published

transmission electron microscopy (TEM) studies (J. Am. Chem. Soc. 2023, 145, 3624; Nano Lett 2020,20, 6582) showing that CrBr₃ crystals grown under conditions nominally identical to those used by us have a higher density of stacking faults and contain parts with different structures, corresponding to different layer stackings. With all this background –and the consistency of all results, including ab-initio calculations –we are convinced that there is compelling evidence that the magnetoconductance jumps originates from different stacking present in CrBr₃ multilayers. We nevertheless agree that this is an assumption and not the result of direct experimental observations.

In the revised manuscript we have changed the phrasing of multiple sentences to avoid giving the impression that we do know the precise structure of the multilayer used and to make clear what are the observations and what is their interpretation, trying to keep the text as factual as possible.

Fig. R2. Raman spectroscopy performed at different positions of two representative bulk crystals at T = 10 K. (a) Raman spectroscopy measurements performed on different positions of a same crystal show exclusively the peaks expected from the thermodynamically stable (ferromagnetic) structural phase. (b) Crystals exhibiting an additional Raman peak at approximately 161 cm⁻¹ can be found in all growth batches, as well as in crystals purchased from commercial providers. We attribute this additional peak to the presence of a considerable amount of a second allotrope of CrBr₃ with a different layer stacking.

Comment 3: 2) Often, the voltage used for the plotted MR or T-dependent resistance data is not given (Fig. 2e-h, Fig. 3a, Fig. 4a-c, Fig. 5, Fig. S1). Please add and, if available comment on the voltage dependence.

Reply 3: We have followed the referee's suggestion by adding the information of the voltage in related figures or figure captions. For a given polarity, changing the voltage bias does not affect the general behaviour of magnetoconductance, and only changes its magnitude. However, for devices with a non-uniform stacking, the qualitative behaviour of the magnetoconductance

is affected by the polarity of the voltage bias (e.g., shown in current Supplemental Fig.5 (Supplemental Fig.3 in previous version), because the dominant contribution to the magnetoconductance originates from the layers closer to the electron injecting contacts, which changes upon changing bias polarity. We have added the related information in the revised Supplementary Information.

Comment 4: 3) *The spot size of the Raman experiments would be helpful and, if it is smaller than the devices, a comment on the homogeneity of the Raman spectra would be adequate.*

Reply 4: We thank the referee for this suggestion. The spot size used in our study was approximately 1 μm . To show the homogeneity of the Raman spectra, we measured different locations on devices (shown in Fig.R3). Our findings indicate that Raman behaviour is consistent and homogeneous across the devices studied. For large bulk crystals, homogeneity over large distances is less good. We have updated the information in our manuscript and supplementary information accordingly.

Fig. R3. Homogeneity of Raman spectra on tunnel barrier devices. Raman spectra measured at three different positions of the device showing (a) ferromagnetic behaviour and (b) L-type antiferromagnetic behaviour.

Comment 5: 4) *There are a few positions, where quotations should be added to substantiate the statement: a) Page 5, line 103: formula for Fowler-Nordheim tunneling b) Page 5, line 106: „magnetic state determines height of the tunnel barrier“ c) Page 5, line 113: „lobes“ ... appear .. due to the divergence of magnetic susceptibility d) page 10, line 262: two of the three expected magnetic states have been observed*

Reply 5: We thank the reviewer for pointing it out and we have added the references.

Comment 6: Minor: 1) *It would be nice to add the thicknesses of the CrBr₃ samples directly to Fig. 3b, e.g. vertically on top. This enables easy direct comparison.*

Reply 6: The referee suggested adding thickness information for all 12 devices in Fig.3b. However, adding this information may cause overcrowding in the figure as there is limited space available. Instead, Supplementary Fig. 2 (previously Supplementary Fig.1) already contains

detailed thickness information. The only issue is that the sample order in Figure 3b and Supplementary Figure 2 are not matched in the previous version, making it difficult to compare the two.

To address this, we have adjusted the order of the devices in Fig.3b of the current version to match the appearance order in the current Supplementary Fig.2. In addition, in the caption of Fig. 3b we refer the reader to Supplementary Fig.2, which contained the sample number and the thickness information.

Comment 7: 2) *Absolute numbers of δG in Fig. 3a would be better. There is no reason to hide this information in the supplement.*

Reply 7: We have changed to absolute numbers of δG in Fig. 3a.

Comment 8 : 3) *The label „8k“ in Fig 1g/h is irritating. At least, in the caption, 8k=8000 should be mentioned.*

Reply 8: We thank the referee for pointing out this problem. We have modified the “8k” to “8000” in related figures.

Comment 9: 4) *I would prefer that all x-axis with out-of-plane H are labelled $H\perp$ as in Fig. 4a. This would make things less ambiguous.*

Reply 9: We have modified the label in related figures as suggested by the referee.

Comment 10 : 5) *Labeling Fig. 1b with AFM(L-type) and AFM(S-type) would make the reading of the paper easier. Else, one has to continuously translate between structural and magnetic labels.*

Reply 10: We thank the referee for this kind suggestion. While Fig. 1b displays the three stacking configurations corresponding to local minima in total energy predicted by first-principles calculations, we believe it would be premature to assign these stackings to specific types at this point, considering the overall logic of the paper. To facilitate the readers' comprehension of the key features of each stacking pattern, we have inserted at a later stage of the manuscript a table.

Table 1 Summary of CrBr ₃ with different magnetic states				
Stacking type	Interlayer magnetic order	Symmetry information	Estimated critical temperature	Magnetoconductance jumps
AA-stacking 	S-type AFM 	$R\bar{3}$	~ 28 K	~ 0.2 T and 0.4 T
M-stacking 	L-type AFM 	$C2/m$	~ 29 K	~ 0.55 T and 1.1 T
AB-stacking 	FM 	$R\bar{3}$	~ 31 K	No jumps

Comment 11 : 6) Two typos: a) page 3, line: 53: ferro and antiferro... b) page 12, line 299: 10°C, 700°C, 580°C

Reply 11: We thank the referee for pointing this out. We have corrected this type of errors in the revised manuscript.

Reviewers' Comments:

Reviewer #2:

Remarks to the Author:

I have read all the review and the responses by the authors. I am satisfied with the responses and recommend publication in Nature Communications.

Reviewer #3:

Remarks to the Author:

The authors' responses are satisfactory. I recommend the manuscript for publication in Nature Communications.

Reviewer #4:

Remarks to the Author:

The remarks of my previous report have been adequately addressed and, as far as I see, also the remarks of the other referees. Consequently, I propose publication without further changes

Reply to Referee #2

Comment: *I have read all the review and the responses by the authors. I am satisfied with the responses and recommend publication in Nature Communications.*

Reply: We thank the referee for the recommendation to publish our manuscript. We appreciate the referee's time and efforts in reviewing our manuscript.

Reply to Reviewer #3:

Comment: *The authors' responses are satisfactory. I recommend the manuscript for publication in Nature Communications.*

Reply: We thank the referee for the recommendation for its publication. We appreciate the referee's time and efforts in reviewing our manuscript.

Reply to Reviewer #4 :

Comment: *The remarks of my previous report have been adequately addressed and, as far as I see, also the remarks of the other referees. Consequently, I propose publication without further changes*

Reply: We thank the referee for the recommendation for its publication. We appreciate the referee's time and efforts in reviewing our manuscript.